# Enhancing tandem mass spectrometry-based metabolite annotation with online chemical labeling

Giovanni Andrea Vitale [1,12], Shu-Ning Xia [1,12], Kai Dührkop [2,3], Mohammad Reza Zare Shahneh [4], Heike Brötz-Oesterhelt [1,5,6], Yvonne Mast [7], Corinna Brungs [8,9], Sebastian Böcker [2], Robin Schmid [8,10], Mingxun Wang [4], Chambers C. Hughes [1,5,6] ✉ & Daniel Petras [6,11] ✉

Metabolite identification in non-targeted mass spectrometry-based metabolomics remains a major challenge due to limited spectral library coverage and difficulties in predicting metabolite fragmentation patterns. Here, we introduce Multiplexed Chemical Metabolomics (MCheM), which employs orthogonal post-column derivatization reactions integrated into a unified mass spectrometry data framework. MCheM generates orthogonal structural information that substantially improves metabolite annotation through in silico spectrum matching and open-modification searches, offering a powerful new toolbox for the structure elucidation of unknown metabolites at scale.

Despite rapid technical advances in high-resolution liquid-chromatography tandem mass spectrometry (LC-MS/MS) hardware and computational methods, metabolite annotation remains a challenging aspect of metabolomic research. On average, less than 10% of features are confidently annotated during MS analysis[1]. While previous studies have suggested that many unannotated features can be attributed to various ion adducts and in-source fragments[2], there is overwhelming evidence estimated by metabolomics[3–6], genome mining[7–9], and natural product discovery data[10–12] that the chemical space of metabolites is vast and mostly uncharted. A critical bottleneck in metabolite identification is the limited coverage of spectral libraries relative to the immense diversity of chemical space[13]. Recent efforts to address this bottleneck include exploiting structural databases, which are substantially larger than spectral libraries, through in silico annotation methods[14], as well as de novo structure annotation methods[15]. Even with advances in in silico annotation, high-confidence annotation of

metabolites is still elusive, motivating the development of workflows to acquire orthogonal chemical data that aid annotation tools and increase accuracy.

Chemical derivatization has been broadly applied to improve structure elucidation and overcome limits of detection by exploiting the unique reactivity of certain chemical moieties[16]. In the field of targeted metabolomics, post-column derivatization is used to enhance the detection of specific compounds[17]. In contrast, derivatization strategies in non-targeted metabolomics have, for the most part, only been successfully implemented as batch processes prior to LC-MS/MS analysis[18–20]. Here, the use of post-column derivatization has enormous untapped potential, as it can be used to provide ion feature-resolved functional group information of unknown molecules. Post-column approaches provide an inherent advantage compared to batch derivatization, as chromatographic co-elution profiles maintain the link between precursors and their derivatization products, which is

¹Department of Microbial Bioactive Compounds, Interfaculty Institute of Microbiology and Infection Medicine (IMIT), University of Tübingen, Tübingen, Germany. ²Chair for Bioinformatics, Institute for Computer Science, Friedrich Schiller University Jena, Jena, Germany. ³Bright Giant GmbH, Jena, Germany. ⁴Department of Computer Science, University of California Riverside, Riverside, CA, USA. ⁵German Center for Infection Research (DZIF), Partner Site Tübingen, Tübingen, Germany. ⁶Cluster of Excellence EXC 2124: Controlling Microbes to Fight Infection, University of Tübingen, Tübingen, Germany. ⁷Department Bioresources for Bioeconomy and Health Research, Leibniz Institute DSMZ - German Collection of Microorganisms and Cell Cultures, Braunschweig, Germany. ⁸Institute of Organic Chemistry and Biochemistry of the Czech Academy of Sciences, Prague, Czech Republic. ⁹Division of Pharmacognosy, Department of Pharmaceutical Sciences, Faculty of Life Sciences, University of Vienna, Vienna, Austria. ¹⁰mzio GmbH, Bremen, Germany. ¹¹Department of Biochemistry, University of California Riverside, Riverside, CA, USA. ¹²These authors contributed equally: Giovanni Andrea Vitale, Shu-Ning Xia. ✉e-mail: chambers.hughes@uni-tuebingen.de; functionalmetabolomics@gmail.com

particularly important for complex mixtures. The gain of structural information in non-targeted chemical metabolomics experiments should increase drastically when orthogonal reactivities are considered. In this case, functional groups can be probed in iterative or parallel reactions, and further multiplexed in one data stream for in silico spectrum annotation and open-modification searches.

Here, we report the development of a Multiplexed Chemical Metabolomics workflow (MCheM) that leverages an array of post-column derivatization reactions for non-targeted LC-MS/MS analysis. To complement the new hardware setup (Fig. 1a), we developed an integrated data analysis pipeline that utilizes the computational concept of ion identity networking in mzmine[21] to post-process MCheM data for downstream metabolite annotation[22]. By integrating CSI:FingerID in silico annotation[23] and GNPS2 open modification search[24] (Fig. 1b), we observed annotation improvements of 31.9% for CSI:FingerID and 37.6% for GNPS2 over an experimental library of ~10 K experimental compounds, and of 48.8% for CSI:FingerID and 20.4% for GNPS2 on a set of authentic natural product (NP) standards. Finally, we demonstrated the utility of MCheM in a genome-guided natural product discovery case study, rapidly identifying

novel oxazolomycin derivatives produced by *Streptomyces libani* subsp. *rufus* DSM 41230.

## Results and discussion
### MCheM hardware and software implementation
To implement MCheM data generation, we designed a custom LC-MS/MS hardware configuration consisting of a make-up UHPLC pump, a T-splitter or reactor manifold, and a syringe pump (Fig. 1a). Using this setup, we implemented three LC-MS-compatible post-column derivatization reactions, each targeting distinct functional groups (Fig. 2a and Supplementary Fig. 1). First, we applied an established derivatization reaction using *L*-cysteine to target electrophiles[18] (Reaction A). Electrophilic groups are fairly common in NPs, as electrophilic NPs often evolved to interact covalently with nucleophilic amino acid residues (e.g., serine, cysteine, threonine) present in biological targets. In our second reaction, we targeted amino and phenol groups using commercially available 6-amino-quinolyl-*N*-hydroxysuccinimidyl carbamate (AQC) (Reaction B). As this reaction requires a basic pH to constrain the amino groups to their free-base form, we infused a 0.5% trimethylamine buffer

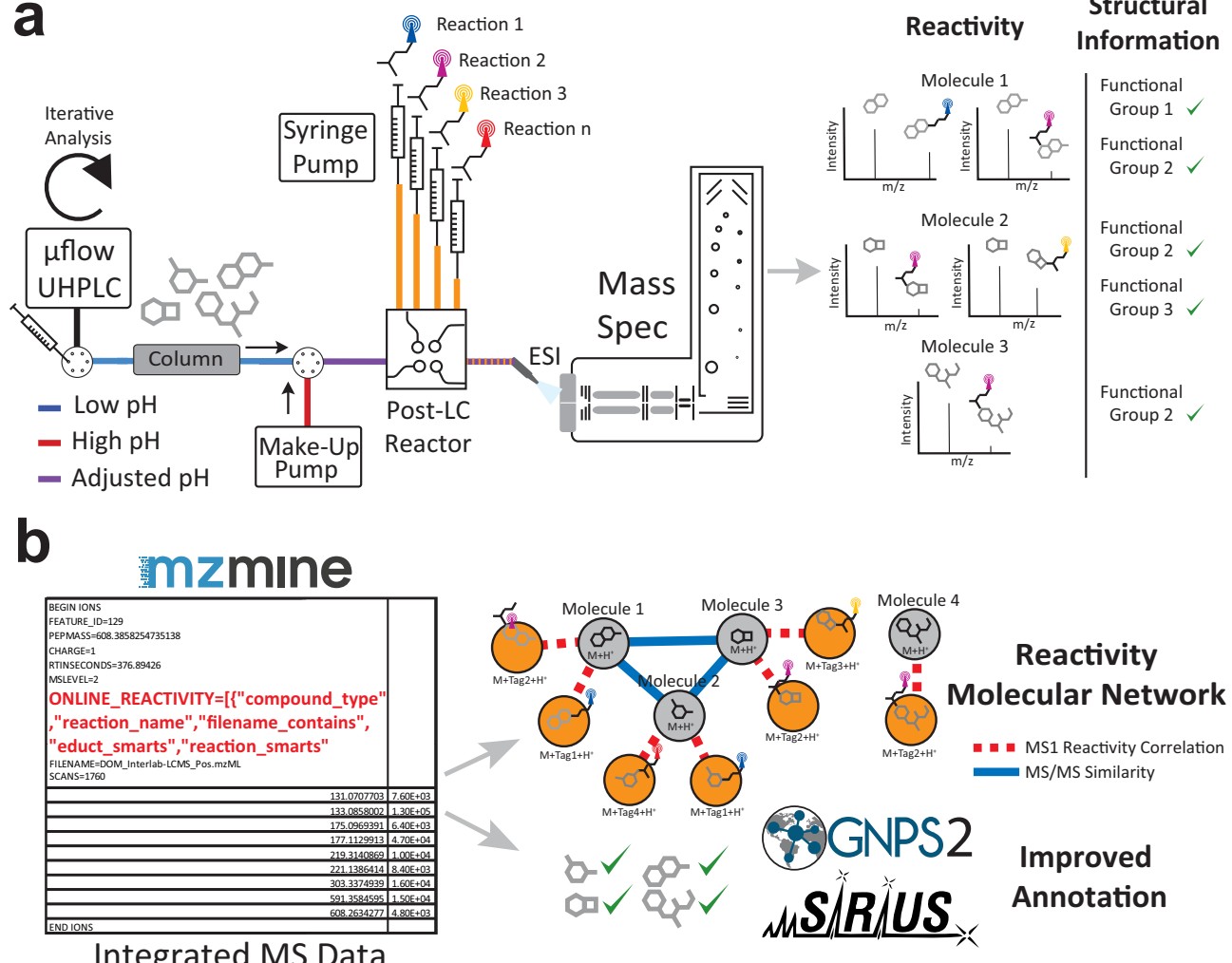

**Fig. 1 | Overview of the Multiplexed Chemical Metabolomics (MCheM) workflow. a** Samples are separated by liquid chromatography (LC), and derivatization agents are continuously infused post-column throughout the run. When necessary (e.g., Reaction B), the effluent pH is adjusted with a buffer infused between the column and the derivatization agent injection port. The effluent is continuously analyzed via MS. **b** The "Online Reactivity" module in mzmine automatically connects precursor/product pairs, integrating structural knowledge into the MS data, and allowing a reactivity molecular network to be visualized. The integrated data are submitted to SIRIUS/CSI:FingerID and GNPS2 to improve annotation confidence.

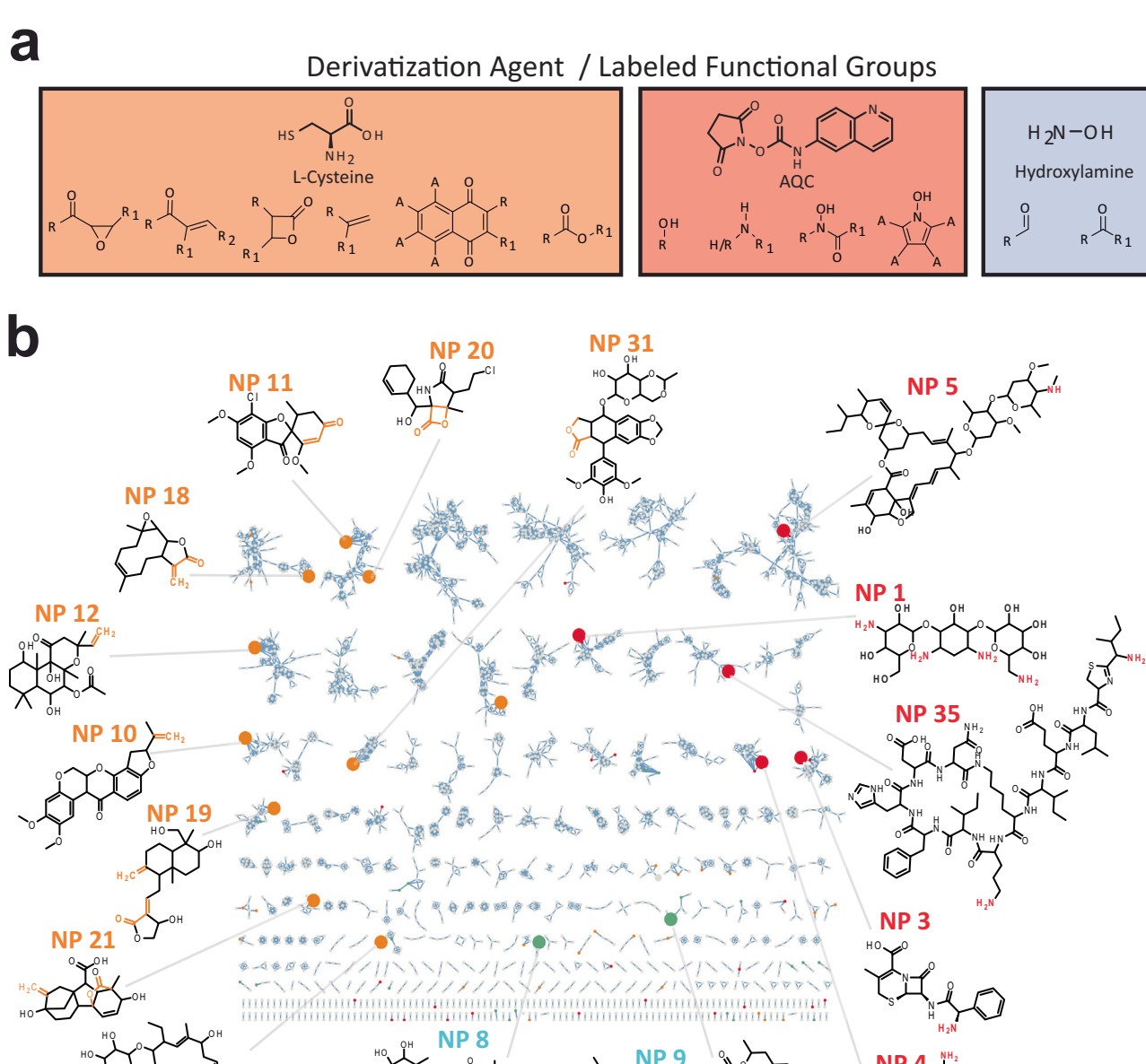

**Fig. 2 | Overview of the Multiplexed Chemical Metabolomics (MCheM) derivatization chemistry and validation with a set of authentic natural product standards. a** Reactions employed in the present study along with the reacting functional groups. **b** Co-clustered reactivity network of a subset of 32 representative natural products here tested, highlighting nodes reacting with cysteine (orange), AQC (red), and *N*-hydroxylamine (light blue). Reacting functional groups are marked with the corresponding colors. Source data is provided through the MassIVE and Zenodo repository (see data availability section and Supplementary Table 2).

between the column and AQC stream using the make-up pump, raising the effluent pH to the 5–6 range during the entire MS analysis. In our third reaction, we targeted aldehydes and ketones using commercially available hydroxylamine hydrochloride (Reaction C)[25]. Following initial test reactions using the three chemical labeling reactions with our post-column reactor setup, we determined concentration-dependent linearity and limits of detection (Supplementary Figs. 2–4). Subsequently, we experimentally validated all three reactions using 359 structurally diverse natural product standards from the Tübingen Natural Compound Collection (https://external.gnps2.org/gnpslibrary) (Supplementary Data 1).

To enable high-throughput analysis of all derivatization reactions in complex metabolomics samples and to merge the resulting data into a single multiplexed data stream, we developed a specialized "Online Reactivity" analysis module in mzmine. This computational strategy leverages the co-elution of precursors and products to establish correlation-based connections (Fig. 1b), using ion identity networking[22] in combination with user-defined Δ*m/z* values corresponding to each derivatization reagent. The resulting MCheM data output represents a hybrid dataset that integrates MS, MS/MS, and reactivity-based information. It also includes a list of predicted functional groups or substructures in the form of SMILES Arbitrary Target

Specification (SMARTS)[26] (Supplementary Figs. 9–11). This reactivity-resolved information can be directly used by downstream computational tools such as CSI:FingerID[14] or GNPS2[24] to constrain the molecular structure search space (Fig. 1b).

## MCheM allows for improved metabolite annotation of tandem mass spectra

We validated the specificity of MCheM reactions using a set of authentic natural product standards, as shown in Fig. 2b. Reaction A successfully labeled electrophilic functional groups, including Michael acceptors[17,27], naphthoquinones[28,29], epoxyketones[18,30], β-lactones[18,31], and macrocyclic esters (likely undergoing thioester formation)[32], as well as terminal alkenes[27]. Reaction B effectively labeled primary and secondary amines, phenols, and N-hydroxy groups[33,34]. Lastly, reaction C labeled aldehydes and ketones[25,35]. A total of 139 distinct derivatization events (including those from Reactions A–C) were detected across the 359 compounds, using their known structures as ground truth. Of these, only five instances (3.6%) were classified as false positives, confirming the high specificity of the MCheM workflow.

We validated the performance of MCheM-enhanced annotation using CSI:FingerID. First, we analyzed 208 spectra from our standard mix that reacted with at least one derivatization reagent, querying them against the SIRIUS biological structure database. For 180 spectra the correct structures were present in the database. The ranking results for every Top k annotation improved due to MCheM, with 88 spectra having their overall rankings improved (49%). Notably, 20% of these spectra were promoted into the top 3, and 6% were reranked to the top 1 position (Fig. 3a). To estimate the improvement in metabolite annotation on a larger and more diverse set of molecules, we

assessed the performance of MCheM by simulating the gained functional group information from 10,709 MS/MS known spectra from MassBank, MoNA and GNPS (CANOPUS dataset[36]), in which we added the SMARTS string to each spectra as ground truth. The corresponding molecular structures were not part of the CSI:FingerID training data. Also here, MCheM substantially improved the annotation rankings for 3297 spectra (32%), with 22% showing improved top 3 and 15% improved top 1 annotations (Fig. 3a). Next, we evaluated the impact of MCheM for open modification search (e.g., in case the target compound is not present in the library, but a structurally highly similar one is present). To do this, we removed the exact matching structures from the GNPS2 MS/MS libraries. Out of the experimental spectra from our 359 authentic standards, 189 yielded at least one hit during open modification spectrum library matching. Using the Tanimoto similarity[37] of molecular fingerprints[38] to evaluate structural similarity, the analysis focused on the queries with a potentially very similar structural analog in the libraries, higher than or equal to 0.5 Tanimoto similarity. This filtration step left 125 MS/MS queries. Comparison of rankings before and after MCheM-informed filtering revealed improvement in the average Tanimoto scores and the rank of the most structurally similar match, as shown in Fig. 3b and Supplementary Figs. 12 and 13. Of the 125 cases, the top 1 Tanimoto score improved in 27 (21.6%) and decreased in 15 (12%), raising the average Tanimoto score from 0.36 to 0.44. Among the top 5 hits, the best Tanimoto scores improved in 28 cases (22.4%) and decreased in 11 (8.8%), raising the average from 0.48 to 0.58. The rank of the highest Tanimoto score improved in 47 cases (37.9%) and declined in 19 (15.3%), with the average rank improving from 14.92 to 9.64 (Supplementary Fig. 16). For the larger CANOPUS dataset, 7248 of the 10,709 public MS/MS spectra yielded analog library matches. Among these, 861 spectra

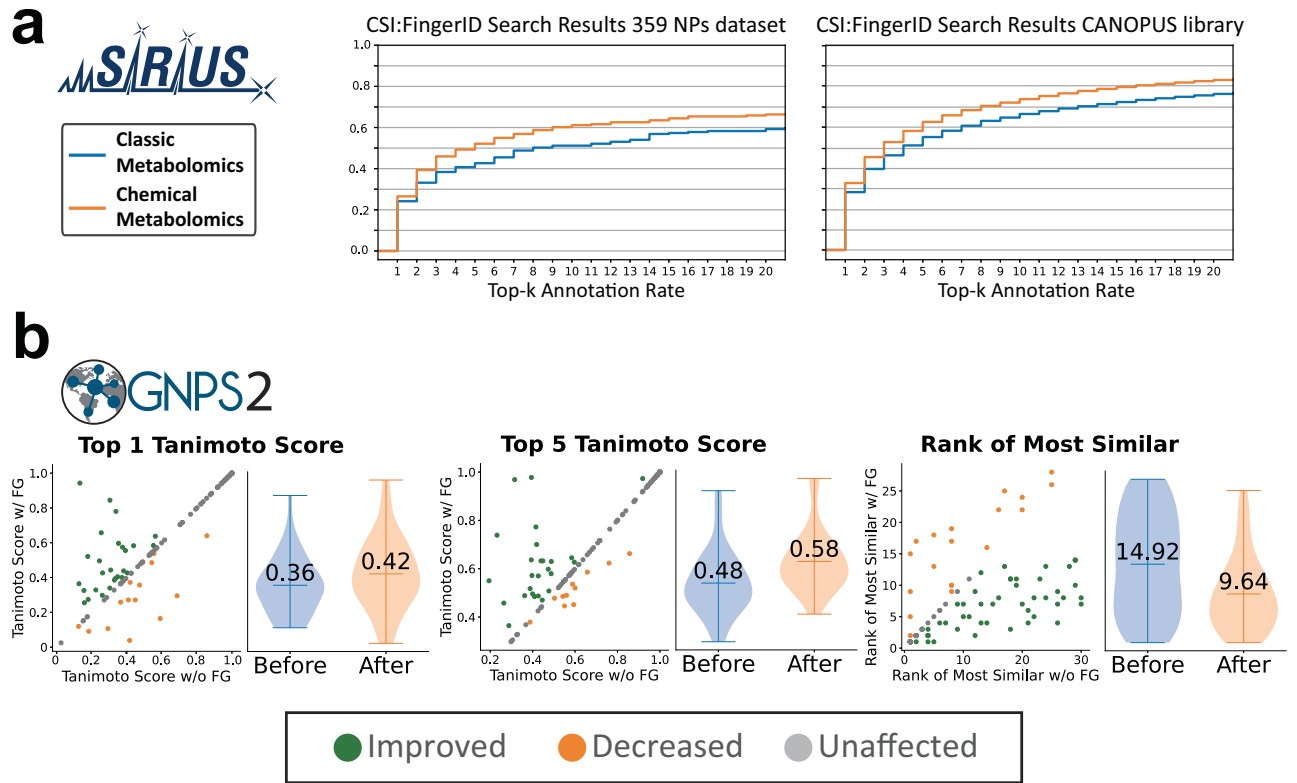

**Fig. 3 | SIRIUS AND GNPS2 results of the Multiplexed Chemical Metabolomics (MCheM) workflow. a** Impact of MCheM on CSI:FingerID search. The annotation rate obtained using only standard MS data (blue) increases when MCheM-enhanced MS data (orange) are used. The improvement is observed for both the experimental dataset and in silico on CANOPUS dataset for every Top k annotation. **b** Impact of MCheM on analog library search. MCheM improved average analogs similarity to the real structure in terms of Top 1 and Top 5 Tanimoto scores, elevating the average Tanimoto score in both cases and for both datasets. Datapoints from the scatter plots are visualized in the violin plots. Error bars indicate the spread of data, and the center line indicates the mean. Source data is provided in the supplemental information.

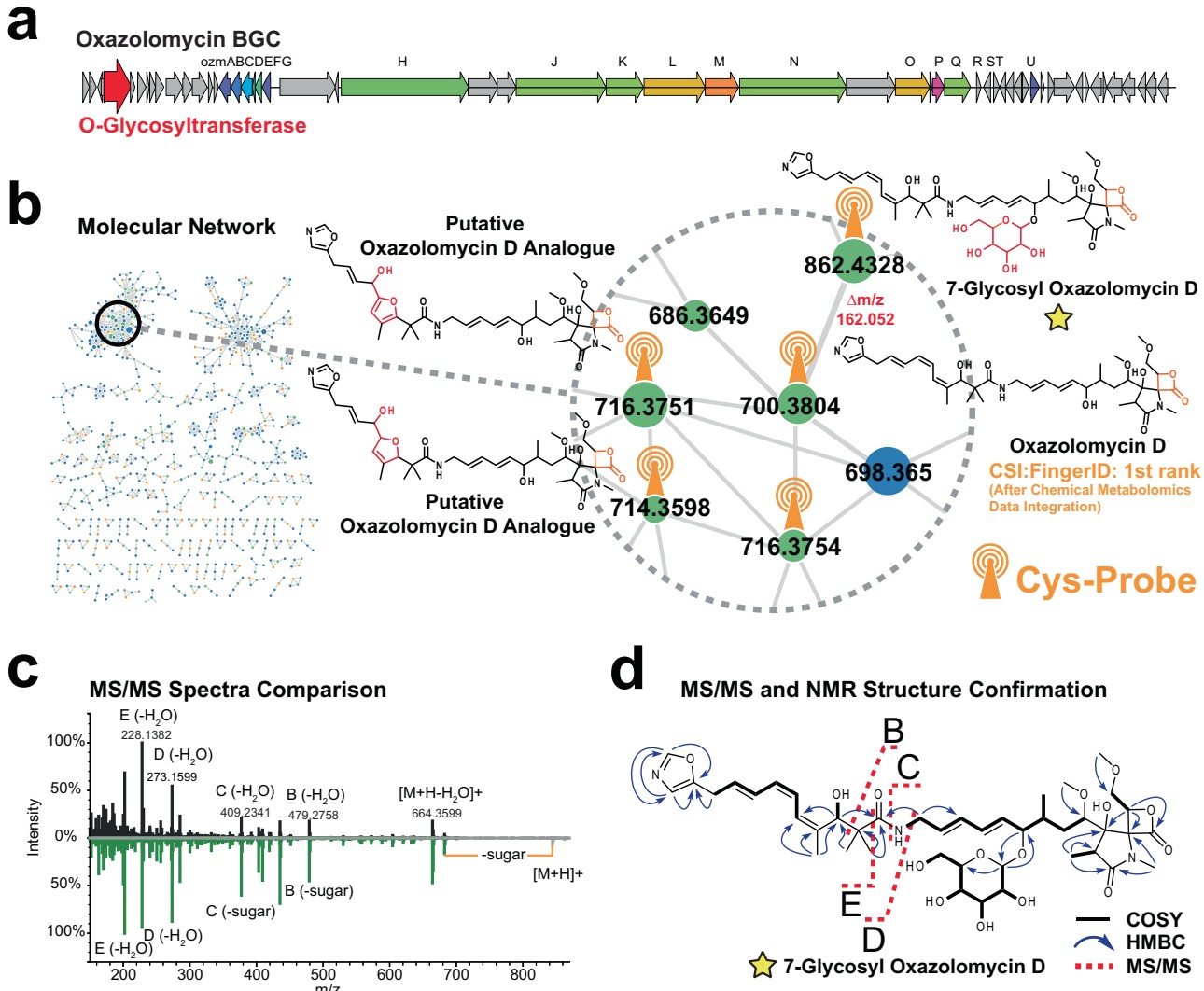

**Fig. 4 | Application of Multiplexed Chemical Metabolomics (MCheM) for the discovery and structure elucidation of 7-glycosyl oxazolomycin D.**
**a** Oxazolomycin biosynthetic gene cluster BGC0001106 from *S. libani* subsp. *rufus* DSM 41230. **b** MCheM reactivity network of *S. libani* extract, with a zoomed view of

oxazolomycin D cluster. **c** Mirror plot of MS/MS spectra of oxazolomycin D and 7-glycosyl oxazolomycin D. **d** NMR and MS/MS structure confirmation of 7-glycosyl oxazolomycin. Source data is provided through the nmrXiv, MassIVE, and Zenodo repositories (see data availability section and Supplementary Table 2).

(11.9%) showed improved top 1 matches, while 562 scans (7.8%) decreased, increasing the average Tanimoto score from 0.44 to 0.52 (an 18% improvement). For the top 5 matches, 887 spectra (12.2%) improved, 446 (6.2%) decreased in ranking, and the average increased from 0.61 to 0.67 (Fig. 3b and Supplementary Figs. 14 and 15). The rank of the most similar matches improved in 2194 cases (30.3%) and worsened in 669 (9.2%), with the average rank improving from 11.94 to 9.42 (Supplementary Fig. 16).

**MCheM facilitates the discovery of 7-glycosyl oxazolomycin D**
Finally, we evaluated the effectiveness of MCheM to explore uncharacterized bacterial extracts. We applied the approach in a genome-guided natural product discovery effort to investigate specialized metabolites produced by *Streptomyces libani* subsp. *rufus* DSM 41230. The *S. libani* genome features a biosynthetic gene cluster (BGC) similar to oxazolomycin B BGC from *Streptomyces albus* JA3453[39] (Fig. 4a). This natural product family features a reactive β-lactone moiety. In order to detect the β-lactone moiety, we leveraged MCheM reaction A. Strikingly, while oxazolomycin was not annotated as one of the top hits using regular MS/MS data, the MCheM-based reranking of CSI:FingerID search results pushed oxazolomycin D to the Top 1

annotation for node *m/z* 700.3804, rendering it the most likely structure (Fig. 4b).

Next, we extended our analysis to explore structurally related, previously uncharacterized derivatives identified through molecular networking and tagged via MCheM reaction A (cysteine). Among the most abundant features within our molecular network were four putative analogs in addition to oxazolomycin D that showed reactivity with the cysteine probe (Fig. 4b). Their mass differences and MS/MS fragmentation patterns indicated possible oxidation and cyclization events in the polyketide chain, consistent with modifications previously reported for oxazolomycin F (Supplementary Figs. 19 and 20)[40]. Another feature with *m/z* 862.434 showed high spectral similarity to oxazolomycin D with a Δ*m/z* of +162.0528 Da (Fig. 4c). This delta mass corresponds to the expected mass of a hexose moiety ($C_6H_{10}O_5$), consistent with the presence of a putative glycosyltransferase gene (srufu_025110) in the *S. libani* oxazolomycin BGC, which is absent in the *S. albus* counterpart (Supplementary Fig. 17). Notably, glycosylated oxazolomycins have not been described in the literature and are not present in natural product databases (GNPS, Dictionary of Natural Products) and other structure databases (e.g., PubChem). Importantly, glycosylated oxazolomycins are not

represented in the structure database searched by CSI:FingerID. To confirm the structure, we purified the compound using flash chromatography and preparative HPLC, followed by a suite of orthogonal NMR experiments (Fig. 4d, Supplementary Figs. 22–27 and Supplementary Table 1). These analyses unambiguously identified the compound as 7-glycosyl oxazolomycin D, representing the first glycosylated member of the oxazolomycin family.

In summary, MCheM combines novel concepts for MS/MS data acquisition and analysis to improve metabolite annotation in non-targeted metabolomics. The workflow includes a scalable software solution in mzmine to detect MCheM-derived features, which can be integrated with downstream MS/MS annotation tools such as SIRIUS and GNPS2 for enhanced metabolite identification. Our results demonstrate that incorporating MCheM-derived reactivity information substantially improves metabolite annotation accuracy, in both in silico and open modification searches, as demonstrated in the natural product discovery of a novel glycosylated oxazolomycin.

A central aspect that underscores MCheM practicality is its simple hardware setup, which can be easily implemented on most commercial LC-MS/MS platforms using a secondary HPLC, a syringe pump, and a readily-available software pipeline. Notably, the improvements reported here were achieved using structural information from only three derivatization reactions. Additional reactions targeting other functional groups can similarly be integrated to further boost annotation. We anticipate that MCheM will be broadly applicable for expanding structural information in non-targeted metabolomics, including the integration of emerging computational approaches with the ultimate goal of providing complete MS/MS-based de novo structure elucidation.

## Methods

### Micro-flow UHPLC-MS/MS

For reactions A and C, LC-MS/MS analysis was performed on a 1290 Infinity II ultrahigh-performance liquid chromatography (UHPLC) system coupled to a Bruker Impact II QTOF mass analyzer. The instrument was operated in data-dependent analysis (DDA) mode with the following parameters: positive ion mode, capillary voltage = 4.5 kV, nebulizer gas pressure = 2.2 bar, dry gas flow rate = 10 L/min, drying temperature = 220 °C. The default mass range was 150–2000 $m/z$ for reaction A and 200–2000 $m/z$ for reaction C, both with a resolution of 50,000. The three most abundant precursor ions were selected for fragmentation with a collision energy (CE) of 20–50 eV. After three MS/MS spectra were acquired on a particular precursor ion, the ion was dynamically excluded from the fragmentation list for 0.3 min. For reaction B, LC-MS/MS analysis was performed on a Vanquish UHPLC equipped with an additional quaternary pump coupled to a Q Exactive HF mass spectrometer. The heated electrospray ionization (HESI) source was used with the auxiliary gas temperature = 400 °C, flow = 12 arbitrary units (AU), sweep gas flow = 1 L/min, sheath gas flow rate = 50 AU. The instrument was operated in data-dependent analysis (DDA) mode with the following parameters: positive ion mode, capillary voltage = 3.5 kV, drying temperature = 250 °C. The default mass range was 220–2000 $m/z$ with a resolution of 120,000. The five most abundant precursor ions were selected for fragmentation with a stepped normalized collision energy (NCE) of 25, 35, and 45 eV, a resolution of 15,000, and an isolation window of 1 $m/z$. The same EVO C-18 column (1.7 μm, 100 Å, 100 ×1 mm) and chromatographic method were employed for all the analyses. The mobile phases consisted of A ($H_2O$ + 0.1% formic acid) and B (ACN + 0.1% formic acid). The constant 150 μL/min flow rate was used following a linear gradient starting with 5% B and reaching 99% B in 10 min, followed by a washing segment between 10 and 12 min (99% B) and a subsequent re-equilibration segment between 13 and 15 min (5% B).

### Chemical metabolomics reactions setup

The derivatization agent (DA) was constantly infused post-column for the whole analysis duration through a syringe pump and a PEEK T-splitter or manifold. Reaction A: L-cysteine, 0.1 mM (10 μL/min), mass range 150–2000 $m/z$. Reaction B: AQC, 100 mM (2 μL/min), mass range 220–1500 $m/z$, with 0.150 mL/min of a 0.5% trimethylamine solution in 1:1 $H_2O$:ACN infused from a second PEEK T-splitter located before the AQC splitter. Reaction C: hydroxylamine hydrochloride, 100 mM (10 μL/min), mass range 200–2000 $m/z$.

### Standard mix

For chemical metabolomics method development, a 100 μg/mL stock mixture containing the following 32 standards in $H_2O$:ACN 1:1 was used: kanamycin (Sigma Aldrich), cephalexin hydrate (Alfa Aesar), doxorubicin hydrochloride (LC laboratories), emamectin (MedChemExpress), vancomycin hydrochloride (Sigma Aldrich), spiramycin (Tokyo Chemical Industry Co.), kitasamycin (AK Scientific), rotenone (MP Biomedicals), griseofulvin (Alfa Aesar), forskolin (LC laboratories), carfilzomib (LC laboratories), tetracycline hydrochloride (Sigma Aldrich), genistein (LC laboratories), erythromycin (Sigma Aldrich), parthenolide (EMD Milipore), andrographolide (Indofine Chemical Company), gibberellic acid (Acros Organics), monensin sodium salt (abcam), yohimbine hydrochloride (Sigma Aldrich), abamectin (abcamBiochemicals), paclitaxel (MedChemExpress), catharanthine (TSZ Chem), harmaline (Indofine Chemical Company), quinine (Sigma Aldrich), vinblastine sulfate (TSZ Chem), etoposide (MedChemExpress), fidaxomicin (ApexBio), mifepristone (Acros Organics), terbinafine hydrochloride (Acros Organics), bacitracin (Sigma Aldrich). Marinopyrrole was obtained via chemical synthesis[41]. Salinosporamide was purified from cultures of *Salinispora tropica*[42].

### Titration experiments

The 32-mix stock (100 μg/mL) was diluted with $H_2O$:ACN 1:1 as a solvent to prepare a 50, 10, 1, and 0.1 μg/mL dilution series. For method B, the 32-mix stock solution was supplemented with L-phenylalanine as internal standard at the same concentration. For the analysis, 5 μL of each standard mix was injected and analyzed in duplicate with and without the chemical metabolomics procedure for each method. The reaction yields were calculated [yield = product area/(product area + educt area) × 100%] for each compound and each reaction and plotted against compound concentrations (Supplementary Figs. 2–4).

**Chemical metabolomics analysis of an in-house natural products library.** A total of 327 pure NPs (from the Tü NP Library) that we recently shared with the metabolomics community via GNPS were mixed into 17 different pools (Supplementary Data 1). For the analysis of each pool, 5 μL were injected and analyzed in duplicates with and without the chemical metabolomics procedure for each experiment, each sample was run in a randomized order. In addition, method A was also evaluated using an extract containing cystargolide A to specifically test the efficiency of β-lactone labeling[43,44]. The extract was obtained by extracting a 6-day old culture of *Kitasatospora cystarginea* NRRL B-16505 grown in KCM liquid medium (1.6 g dextrin, 0.8 g galactose, 0.8 g maltose, 0.8 g Bacto soytone, 0.4 g glucose, 0.3 g $(NH_4)_2SO_4$ dissolved in 1 L tap water) with ethyl acetate.

**Data processing.** Thermo raw data were converted into .mzML format using msConvert by Proteowizard[45], while Bruker raw data were converted into the same format through a script developed by the manufacturer. All the MS data were processed with mzmine (Version 4.1.0) to "clean up" and align the data, and reactive metabolites were automatically annotated through the "Online Reactivity" module. Here different theoretical Δ$m/z$ values were set according to the expected products for each derivatization reaction: 121.0197 Da for reaction A, 170.0481 Da for reaction B, and 15.0109 Da for reaction C (Supplementary Figs. 5–7 and Supplementary Data 2). All metabolites were annotated, and all possible reactive substructures present across the metabolites were incorporated into a table in SMARTS format and

automatically integrated into the.mgf files for SIRIUS and GNPS (Supplementary Data 2). The correct SMARTS representation was tested using a script written in Python. Reacting metabolites but lacking any possible reactive substructure were listed as "false positives". Metabolites with a low conversion yield (<5%) in reaction A were excluded to refine the data. The parameters employed for processing each dataset and the output data are publicly accessible as mzmine batch files (Supplementary Table 2). The data were exported with the FBMN module as one file containing the MS and MS/MS features (.mgf), one quantification table (.csv), and an edges annotation table (.csv), and through the SIRIUS exporting module as one MS file (.mgf).

**CANOPUS dataset and CSI:FingerID evaluation.** The *CANOPUS dataset* consists of 8,569 tandem mass spectra from GNPS[24], 1417 tandem mass spectra from MoNA (https://mona.fiehnlab.ucdavis.edu), and 723 tandem mass spectra from MassBank[46], containing a total of 8553 unique structures[36]. For this dataset, results of the four MCheM reactions were determined using the known compound structures, assuming error-free chemistry. Annotations were performed using the latest version of CSI:FingerID that is part of SIRIUS 6, using a *worst-case* five-fold cross validation: the fingerprint for each spectrum was predicted using a machine learning model that has seen neither any spectrum of the same structure, nor any spectrum of a similar or derivatized structure. Chemically similarity was determined using the myopic Maximum Common Edge Subgraph distance[47].

**GNPS2 data analysis.** An analog library search[24] was conducted to retrieve the top 30 analogs for each reading. The search parameters were adjusted to be more relaxed than the GNPS default (minimum cosine = 0.5, minimum matched peaks = 4, fragment ion tolerance = 0.5, and precursor ion tolerance = 2.0) (https://gnps2.org/homepage). This adjustment leverages additional FG information to capture more structurally similar matches that might exhibit lower cosine values. For the analysis section of this work, duplicate hits and exact matches were removed from the library search results, and the search was expanded to maintain a complete set of analogs. This step ensures that the results consist only of analogs and not exact library matches, ensuring fair comparisons and eliminating potential information overlap. It should be noted that the remaining structures in the cleaned libraries may still show a Tanimoto similarity of 1 with some data structures, as the molecular fingerprint representation is lossy, meaning two structurally distinct molecules can produce the same fingerprint. The fingerprints were computed using the default settings of the RDKit (https://www.rdkit.org) fingerprint implementation. Next, the Tanimoto similarity between the fingerprint of each reading's structure and the fingerprints of its matches were calculated as the baseline measure of success. The matches were then ranked based on their modified cosine similarity score, referred to as the "ranking without functional groups." Finally, the structures of the matches were examined to determine if they contain the corresponding functional groups. With this information, the "ranking with functional groups" was calculated, where matches are first ordered by the presence of the functional group and then by their modified cosine score.

**Genome analysis of DSM 41230.** The genome sequence of *S. libani* subsp. *rufus* DSM 41230 (taxonomic correct name *Streptomyces platensis* DSM 41230) was analyzed with antiSMASH v. 7.0[48] to identify BGCs. Region 1.19 contained a set of genes similar to the oxazolomycin B BGC from *S. albus* JA3453. The analysis of synteny among clusters was performed with clinker[49].

**Isolation and structure elucidation of 7-glycosyl oxazolomycin D.** Strain *S. libani* subsp. *rufus* DSM 41230 was first cultivated in a 100 mL Erlenmeyer flask with 30 mL of R5 liquid medium (100 g sucrose, 10 g

glucose, 0.25 g $K_2SO_4$, 10 g $MgCl_2$, 0.1 g casamino acids, 5 g yeast extract, 5.7 g TES buffer, and 2 mL trace metal mix in 1 L distilled water, pH 7.2). The culture was grown on a platform shaker at 100 rpm and 29 °C for three days. 5 mL of preculture was used to inoculate 500 mL flasks containing 150 mL fermentation medium (4 g yeast extract, 10 g soluble starch, 2 g Bacto peptone and 35 g sea salts in 1 L distilled water, pH 7.0). The culture was grown at 100 rpm and 29 °C for four days. The complete culture (10.5 L) was then centrifuged, and the supernatant was extracted three times with an equal volume of ethyl acetate (EtOAc). The EtOAc extract was dried over $NaSO_4$ and concentrated under reduced pressure to give 6.7 g of crude extract. The extract was fractionated on silica gel using a stepwise gradient elution with the following solvents: 100% hexanes, 30% EtOAc in hexanes, 100% EtOAc, 1% MeOH in DCM, 10% MeOH in DCM, 100% MeOH. Fraction 5 (eluted with 10% MeOH in DCM) was dried and subjected to further silica gel chromatography using the following elution profile: 100% DCM, 10% MeOH in DCM, and a linear gradient from 10-100% MeOH in DCM over 20 min, yielding multiple subfractions. Fractions containing the target compound were combined, dried, and redissolved in 7% MeOH in DCM for preparative HPLC purification (Phenomenex Luna silica (2), 10 μm, 250 × 21.2 mm) using an isocratic method (7% MeOH in DCM, 10 mL/min) and UV detection at 254 nm to yield 7-glycosyl oxazolomycin D (53.4 mg, $t_R$ = 12 min): HRESIMS $m/z$ $[M + H]^+$ = 862.4330, calculated for $C_{43}H_{64}N_3O_{15}^+$, 862.4332. The compound was characterized 1D and 2D NMR spectroscopy, including $^1H$, $^{13}C$, COSY, HSQC, HMBC, NOESY experiments, recorded on a Bruker Avance III HDX 700 MHz spectrometer equipped with a 5 mm Prodigy ($^1H$,$^{19}F$/$^{13}C$/$^{15}N$) TCI Cryoprobe (Supplementary Table 1 and Supplementary Figs. 22–27). $^1H$ NMR data were recorded at 700 MHz in DMSO-$d_6$ (2.50 ppm), and $^{13}C$ NMR data were recorded at 175 MHz in DMSO-$d_6$ (39.5 ppm). NMR data were processed using MestReNova (Mnova 14.3.0, Mestrelab Research) software[50].

## Reporting summary

Further information on research design is available in the Nature Portfolio Reporting Summary linked to this article.

## Data availability

All MS data acquired in this work as well as processed data, and processing batch files, are publicly available through both MASSIVE and Zenodo [https://zenodo.org]. All accession numbers and links are listed in the supplemental information in Supplementary Table 2. The CANOPUS dataset used to test the method can be downloaded from [https://bio.informatik.uni-jena.de/data/]. *Streptomyces libani* subsp. *rufus* DSM 41230 (strain NBRC 15424) genome is publicly accessible through NCBI Nucelotide Database with accession number AP023408. The GNPS2 FBMN job is publicly available at the following link: [https://gnps2.org/status?task=714b3ae18f8d4c499876d486658b9b30]. GNPS2 Chemical Metabolomics job for the experimental dataset can be accessed here: [https://gnps2.org/status?task=508a9676780d4e119cb6ac3bce011512]. NMR data has been deposited to nmrXiv with the https://doi.org/10.57992/NMRXIV.P114. Source data are provided with this paper.

## Code availability

The code for the GNPS workflow and analysis is available on GitHub [https://github.com/Wang-Bioinformatics-Lab/Chemical_Metabolomics_with_functional_groups/tree/master] and Zenodo [https://zenodo.org/records/15469092]. A fully runnable instance of the workflow can be launched directly at GNPS2.org [https://gnps2.org/workflowinput?workflowname=Chemical_Metabolomics_With_Functional_Groups]. The mzmine source code and versions are available on GitHub [https://github.com/mzmine/mzmine] and mzio.io [https://mzio.io]. MCheM filtering is available in SIRIUS version 6.2, and source code and versions are available on GitHub [https://github.com/sirius-ms/sirius] and brightgiant.com [https://bright-giant.com]. The code for filtering SIRIUS

candidate lists via MCheM can be found at GitHub [https://github.com/kaibioinfo/functional_metabolomics]. The code for the SMARTS check is uploaded to GitHub [https://github.com/corinnabrungs/smarts_testing] and Zenodo [https://doi.org/10.5281/zenodo.15480298].

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

## Acknowledgements

We would like to thank Markus Fleischhauer (Bright Giant GmbH) for assistance with the MCheM code implementation in SIRIUS. This study was supported by the Deutsche Forschungsgemeinschaft (German Research Foundation, DFG) via the Cluster of Excellence EXC 2124: Controlling Microbes to Fight Infection (CMFI, project ID 390838134) to H.B.O., C.C.H., and D.P. and via TRR 261 (project ID 398967434) to H.B.O. and D.P., and a research grant to K.D. and S.B. (BO 1910/23). D.P. was supported by the Simons Foundation through an Simons Early Career Investigator in Aquatic Microbial Ecology and Evolution Award (SFI-LS-ECIAMEE-00013858). M.W. was supported by NIH 5U24DK133658-02 and by the U.S. Department of Energy Joint Genome Institute (https://ror.org/04xm1d337), a DOE Office of Science User Facility, supported by the Office of Science of the U.S. Department of Energy operated under Contract No. DE-AC02-05CH11231. Additional funding was provided by the Deutsches Zentrum für Infektionsforschung (German Center for Infection Research, DZIF) to H.B.O., C.C.H., Y.M., project TTU 09.826. S.X. is grateful for a PhD scholarship (202008330294) from the Chinese Scholarship Council. C.B. was supported by the Czech Academy of Sciences PPLZ fellowship number L200552251. The mzmine project is funded by the European Union, the BAB - Funding Bank for Bremen and Bremerhaven, and the Senator of Economics, Ports and Transformation Bremen (65002459).

## Author contributions

C.C.H. and D.P. conceptualized the MCheM approach and supervised the study. G.A.V., C.C.H. and D.P., performed the derivatization reactions and LC-MS/MS experiments. K.D., M.Z.S.H., C.B., S.B., R.S., and M.W. developed software. G.A.V., K.D., M.Z.S.H. and D.P. performed MCheM data analysis. H.B.O. and Y.M. provided natural product standards and strains. Y.M. performed BGC analysis. S.N.X. and C.C.H. performed natural product purification and NMR experiments. G.A.V., C.C.H. and D.P. wrote the manuscript. All authors edited and approved the manuscript.

## Competing interests

M.W. is a co-founder of Ometa labs LLC. R.S. is a co-founder of mzio GmbH (Bremen, Germany). S.B. and K.D. are co-founders of Bright Giant GmbH (Jena, Germany). The remaining authors declare no competing interests.
