## [Transparent Peer Review file · Nature Communications]

Enhancing tandem mass spectrometry-based metabolite annotation with online chemical labeling

Corresponding Author: Professor Daniel Petras

Version 0:

Reviewer comments:

Reviewer #1

(Remarks to the Author)

The manuscript addresses one of the most significant challenges in untargeted metabolomics. Our research group has been grappling with this issue, often struggling to annotate even 1% of the significant features identified. Here the authors aim to develop a multiplexed chemical metabolomics (MChem) workflow that leverages a series of post-column derivatization reactions for untargeted LC-MS/MS analysis. Three distinct types of derivatization reactions were employed, each tailored to specific groups of compounds. This innovative approach successfully facilitated molecular annotation.

Beyond the hardware setup, the authors designed an integrated data analysis pipeline using the computational concept of ion identity networking in MZmine to process MChem data, enhancing the efficiency of metabolite annotation. The workflow was validated using an experimental library of 10,000 compounds and authentic standards. Additionally, it was applied to uncharacterized bacterial extracts, leading to the first-ever annotation of glycosyl oxazolomycin D, which was confirmed via NMR.

This manuscript demonstrates high scientific quality and is anticipated to make valuable contributions to the field of metabolomics. All MS experiments were meticulously conducted, and the data is publicly available, enabling readers to thoroughly evaluate the findings. Moreover, the supplemental material is exceptionally rich and should be explored by readers.

(Remarks on code availability)

Reviewer #2

(Remarks to the Author)

The manuscript describes a new framework of mass spectrometry (MS)-based metabolomics. Non-targeted MS-based metabolomics still include difficulties to identify each secondary metabolite. The authors integrated orthogonal post-column derivatization reactions into the system and consequently improved annotation of unknown metabolites.

The framework is a new and powerful. The usefulness is well explained in the manuscript. As a proof of concept, new oxazolomycin congeners were also discovered from culture extract of a *Streptomyces* strain. Thus, this work is worth publishing. As the manuscript is perfect, there is no comment that the authors need to address.

(Remarks on code availability)

Reviewer #3

(Remarks to the Author)

Very interesting paper on a timely topic. The work is the most innovative paper that I have read in my 40 years in research, publishing and editing Journals, and in my roles I see more than 300 annually.

The topic has great impact to various health-agrofood research innovation works. Metabolite is the bottleneck and a source of errors. Improving its accuracy and productivity is of utmost importance.

If this is wrong then all the produced biochemical theories are wrong.

The present work combines strong novelty in the following fields: derivatisation reaction, analytical chemistry, data mining and the related searches in software and databases also implementing new intuitive hardware solution.

(Remarks on code availability)
